# Hypoglycemic Effects and Mechanisms of Buckwheat–Oat–Pea Composite Flour in Diabetic Rats

**DOI:** 10.3390/foods11233938

**Published:** 2022-12-06

**Authors:** Xueqian Yin, Siqi Liu, Xiaoxuan Zhang, Yuanzhi Jian, Jing Wen, Ruoyu Zhou, Ning Yin, Xinran Liu, Chao Hou, Junbo Wang

**Affiliations:** Department of Nutrition and Food Hygiene, School of Public Health, Peking University, Beijing 100191, China

**Keywords:** diet intervention, diabetes mellitus, rats, buckwheat, oats, peas, lipids, liver, gut microbiota

## Abstract

Nutritional intervention is a basic way to prevent and treat diabetes mellitus. Appropriate whole grain intake daily is recommended. The study aimed to explore the feasibility of a kind of buckwheat–oat–pea composite flour (BOP, quality ratio of buckwheat:oats:peas = 6:1:1) as a stable food substitution and its underlying mechanisms. High-fat food (HFD) and streptozotocin injection were used to induce diabetes in rats, and buckwheat, oats, and three different doses of BOP were added to the HFD separately for diet intervention. The whole study lasted for 10 weeks, and the glucose tolerance test, lipids, liver injury, and gut microbiota were evaluated in the last week. The diabetic rat model was successfully induced. The BOP significantly changed the glucose and lipids metabolism, decreased liver injury, and changed the composition of the gut microbiota of diabetic rats. The outcomes of the current study revealed that BOP is a potential stable food substitution.

## 1. Introduction

The epidemic of diabetes mellitus is a major threat to global health. Diabetes and its complications are the major cause of blindness, kidney failure, heart attacks, stroke, and lower limb amputation [1]. According to the International Diabetes Federation, 1 in 10 adults aged 20–79 years had diabetes in 2021, and it is estimated that there will be more than 643 million diabetic people by 2030 [2]. Nutrition therapy plays an important role in diabetes management. According to the latest recommendation of the American Diabetes Association, carbohydrates should be sourced from high-fiber products, and whole grains should be emphasized in the diet [3].

With high fiber, a complementary amino acid pattern with rice and wheat, some special functional phytochemicals, coarse cereals have attracted the attention of nutritionists. For example, buckwheat [4] and its flavonoids, such as rutin [5] and quercetin [6], oats [7], and β-glucan [8,9], showed the effect of decreasing blood glucose according to some animal experiments and clinical trials. Peas were also found to have some beneficial effects in glucose-intolerant rats [10]. As coarse cereals become popular, some problems have arisen. Poor tasting and hard processing characteristics inhibit coarse cereal consumption. However, mixing several coarse cereals together is a simple and low-cost way to partially solve these problems. For example, peas could change the color, lightness, and viscosity of flour when mixed with buckwheat [11], and the bread made from oat, rye, buckwheat, and wheat composite flour had high palatability and processability [12]. As the phytochemicals may have synergetic or antagonistic effects, the best mixing pattern needs to be studied.

In our previous study, we found that a mixture of buckwheat, oats, and peas (weight ratio = 6:1:1) could decrease insulin resistance in vitro. Briefly, simulated gastrointestinal digestion products of buckwheat, oats, peas, and coix were added to insulin-resistant HepG2 cells to find those that could improve glucose consumption. The effects rank order was buckwheat > oats > peas > coix, so the buckwheat and oats were selected for primary ratio selection. When the ratio of buckwheat to oats was 3, the mixture had a maximum effect of improving glucose consumption. However, considering the nutritional and processing value, peas were added. Then, we tested different mixtures in which 25%, 50%, and 75% of oats were replaced by peas, respectively, in vitro glucose consumption testing. The buckwheat:oats:peas = 6:1:1 pattern was found to be the best one (Appendix A). Then, the pattern was tested in vivo and was found to have the effects of decreasing blood glucose and diabetic symptoms [13]. This study aimed to explore the underlying mechanisms of the hypoglycemic effects, including lipid metabolism, liver functioning, and gut microbiota.

## 2. Materials and Methods

### 2.1. Preparation of Animal Diets

Buckwheat (*Fagopyrum esculentum* Moench, Dasanleng), oats (*Avena nuda* L., Bayou-1), and peas (*Pisum sativum* L., Zhongwan), which were all purchased from Dongfangliang Life Technology Co., Ltd. (Datong, China), were dried, ground, and then mixed at a quality ratio of 6:1:1 to make buckwheat–oat–pea composite flour (BOP).

Animal diets came into two categories: normal diet (AIN-93 M diet, maintenance diet recommended by the American Institute of Nutrition [14]) and high-fat diet (HFD, 45% calories from fat). Then, buckwheat flour (10%), oat flour (10%), and BOP (3.3%, 10%, 30%) were separately added to the HFD to obtain 5 different diets. At the same time, the proportion of macronutrients was adjusted so that the amounts of carbohydrate, protein, and fat in the HFD remained the same (Table 1). All diets were processed by BiotechHD Co., Ltd. (Beijing, China).

### 2.2. Animal and Housing Environment

Male Sprague Dawley (SD) rats (180 ± 20 g) were purchased from the Department of Laboratory Animal Science of Peking University (Beijing, China) and were housed in a specific pathogen-free room with a controlled temperature (25 ± 1 °C), relative air humidity (50~60%) and 12 h light/12 h dark cycles. The rats had free access to the diet and water. This experiment was reviewed and approved by the Ethics Committee of Peking University (ethics no. LA2019362).

### 2.3. Establishment of the Diabetic Rat Model and Experimental Treatments

After a 7-day adaptive period, 64 rats were randomly divided into 8 groups by fasting blood glucose (FBG) and body weight, including the normal control (NC) group, model control (MC) group, buckwheat (BU) group, oat (OA) group, metformin (MET) group and three BOP groups (low-dose (BOP-L) group, medium-dose (BOP-M) group, and high-dose (BOP-H) group) (Figure 1). The rats in the NC group were fed with the normal diet, while the others were fed with the HFD. However, the HFD of the BU, OA, BOP-L, BOP-M, and BOP-H groups contained 10% buckwheat flour, 10% oat flour, 3.3% BOP, 10% BOP, and 30% BOP, respectively.

After 30 days, apart from the NC group, the rats in the other groups received 30 mg/kg streptozotocin (Sigma Chemical Co., Ltd., St Louis, MO, USA) intraperitoneal injection twice to induce diabetes, with an interval of 7 days. The streptozotocin was dissolved in 0.1 mol/L (PH = 4.5) citric acid sodium citrate buffer (Bioroyee Biotechnology Co., Ltd., Beijing, China) when used. On the 5th day after the last injection, an oral glucose tolerance test was performed, and if the fasting blood glucose (FBG) and the area under the blood glucose curve (GAUC) in the oral glucose tolerance tests (OGTTs) of the rats in the MC group were significantly higher than those of the NC group, then the model was regarded to be successful. Then, the rats in the MET group were given metformin hydrochloride (Sino-American Shanghai Squibb Pharmaceuticals Ltd., Shanghai, China) by oral gavage every day (the period of gavage was 4 weeks, according to the drug instructions, the doses were 100 mg/kg/d, 150 mg/kg/d, 200 mg/kg/d, and 200 mg/kg/d from the 1st to the 4th week), and the other groups were given the same dose of distilled water at the same time. An OGTT was conducted in the last week.

Finally, blood samples were taken from the femoral artery and centrifuged at 3000× *g* for 10 min for serum extraction after 12 h of fasting. The caeca contents were collected, and all the samples were snap-frozen in liquid nitrogen and stored at −80 °C until further analyses. Sections of liver tissue were fixed with formalin for histological analyses.

### 2.4. Oral Glucose Tolerance Test

After fasting for 5 h (8:00–13:00), the rats were given a solution of glucose (2.0 g/kg body weight) by oral gavage. The blood glucose levels in the tail were measured before (0 min) and 30 min, 60 min, and 120 min after glucose gavage by the glucose oxidase method using a blood sugar meter (On Call Plus, ACON Biotech Co., Ltd., Hangzhou, China). The GAUC was calculated by the formula below:

GAUC = (0 h blood glucose + 0.5 h blood glucose × 2+ 1 h blood glucose × 3 + 2 h blood glucose × 2)/4.

### 2.5. Serum Biochemical Analysis

The levels of total cholesterol (TC), triglyceride (TG), low-density lipoprotein cholesterol (LDL-C), high-density lipoprotein cholesterol (HDL-C), alanine aminotransferase (ALT), and aspartate aminotransferase (AST) in serum were detected using an AU480 automatic biochemistry analyzer (Beckman Coulter, Inc., Brea, CA, USA). TC was measured by cholesterol oxidase phenol 4-aminoantipyrine peroxidase method, TG was measured by glycerol phosphate oxidase-p-aminophenazone method, LDL-C and HDL-C were measured by direct method, and ALT and AST were measured by substrate method. All kits were purchased from Autobio Diagnostics Co., Ltd. (Beijing, China).

### 2.6. Histological Analysis

Liver tissues were randomly selected from 3 rats in every group. After being fixed in 10% neutral buffered formalin for 24 h [15], the tissues were embedded in paraffin, sectioned to 5 μm, and finally stained with H&E. The slides were observed under a Nikon E400 light microscope (Nikon Instruments (Shanghai) Co., Ltd., Shanghai, China), and pictures were taken at 200×.

### 2.7. Gut Microbiota Analysis with 16S rDNA Gene

Caeca contents were collected randomly from 5 rats in each group. Microbial DNA was extracted using the E.Z.N.A.^®^ soil DNA kit (Omega Bio-Tek, Inc., Norcross, GA, USA) according to the manufacturer’s instructions. The V3–V4 hypervariable regions of the bacterial 16S rRNA gene were amplified with universal primers (338F 5′-ACTCCTACGGGAGGCAGCAG-3′, 806R 5′-GGACTACHVGGGTWTCTAAT-3′) by an ABI GeneAmp^®^9700 thermocycler PCR system (Thermo Fisher Scientific Inc., Waltham, MA, USA). The PCR reactions were conducted using the following program: 3 min for denaturation at 95 °C, 27 cycles of 30 s at 95 °C, 30 s for annealing at 55 °C, and 45 s for elongation at 72 °C, followed by a final extension at 72 °C for 10 min. PCR reactions were performed in triplicate 20 μL mixture containing 4 μL of 5 × FastPfu Buffer, 2 μL of 2.5 mM deoxyribose nucleoside triphosphates, 0.8 μL of each primer (5 μM), 0.4 μL of FastPfu Polymerase (TransGen Biotech Co., Ltd., Beijing, China), and 10 ng of template DNA. After being extracted and purified using an AxyPrep DNA gel extraction kit (Corning Inc., New York City, NY, USA), amplicons were pooled in equimolar amounts and paired-end sequenced on an Illumina MiSeq platform (Illumina Inc., San Diego, CA, USA). Raw data were quality filtered and trimmed, denoised, and merged, and then the chimeric sequences were identified and removed to obtain the feature table of amplicon sequence variants (ASVs) by QIIME2. The ASV table was aligned with a pretrained 99% similarity GREENGENES 13_8 database with a threshold of 70% to generate the taxonomy table. Then, the mitochondrial and chloroplast sequences were removed, and the representative sequences of operational taxonomic units (OTUs) were obtained.

### 2.8. Statistical Analysis

Statistical analysis was performed using R 4.0.2 (R Foundation for Statistical Computing, Vienna, Austria). Normally distributed variables were presented as the mean ± SD, whereas nonnormally distributed variables were expressed as the median (25% interquartile range (IQR), 75% IQR). Shapiro–Wilks test was used to test the normality of data. Two-sample Student’s t-test and one-way analysis of variance (ANOVA) were used to identify differences between groups of normally distributed variables. When ANOVA found significant differences, multiple comparisons among the MC group and the other groups, three BOP groups, and the BU group or the OA group were conducted. The Welch test, variable transformation, or Kruskal–Wallis test was used when necessary. For gut microbiota data, the Kruskal–Wallis test and permutational multivariate analysis of variance (PERMANOVA) test were used to analyze the alpha diversity and beta diversity among groups, respectively. Deseq2 analysis was conducted to find the differences in the distribution of bacteria between each of the two groups. The “FDR” method was used for I error control. Differences were considered statistically significant at α < 0.05.

## 3. Results

### 3.1. Establishment of the Diabetic Model and Hypoglycemic Effects of BOP

The diabetic model was successfully induced after streptozotocin injection: the FBG (NC group vs. MC group: 4.5 (4.18, 4.73) mmol/L vs. 12.1 (11.2, 13.7) mmol/L, *p* < 0.001) and GAUC (NC group vs. MC group: 11.58 ± 0.95 vs. 44.84 ± 6.50, *p* < 0.001) of the rats in the MC group were significantly higher than those in the NC group, as have been shown in our previous study [13]. The blood glucose curve of OGTT of the NC group in the last week raised in 0 to 60 min, arrived at glucose peak at 60 min, and declined then, while the curves in the diabetic rats (except for the MET group) followed a different shape: the glucose peak arrived at 30 min. Further, curves of the other groups were all below the MC group, and the BOP-L and BOP-M groups showed lower glucose at 0 min (*p* < 0.05), while the BU and OA groups also had lower glucose levels at 0 and 60 min compared with the MC group (*p* < 0.05); however, there was no significant difference when comparing the three BOP groups with the BU group or the OA group in every time point (*p* > 0.05), as shown in Figure 2.

### 3.2. Effects of BOP on Blood Lipids

Compared with the NC group, the levels of TC, TG, HDL-C, and LDL-C in the MC group all significantly increased (*p* < 0.05). Compared with the MC group, the BOP-L, BU, and MET groups all had significantly lower TC and HDL-C levels; the BU and MET groups also had lower TG levels (*p* < 0.05). There was no significant difference between the three BOP groups and the BU group or the OA group (Table 2).

### 3.3. Effects of BOP on Liver Injury

Serum hepatic enzymes such as ALT and AST are common markers of liver injury. The MC group had higher serum ALT and AST levels than the NC group (*p* < 0.05), indicating that the diabetic rats had liver injury. Compared with the MC group, all intervention groups (MET, BOP-L, BOP-M, BOP-H, BU, OA) had significantly lower ALT levels (*p* < 0.05), while the MET, BU, OA, BOP-H groups also had lower AST levels (*p* < 0.05). Additionally, there was no significant difference in the serum ALT and AST levels between the three BOP groups and the BU group or the OA group (*p* > 0.05) (Figure 3a,b).

Similar results could be seen in the histological analysis. Compared with the NC group, the hepatocytes in the MC group became larger and had ballooning and microvesicular steatosis, the cell boundaries became unclear, and the hepatic sinusoids became narrower. After the intervention, balloon degeneration decreased, and the cell boundaries became clear. The nonalcoholic fatty liver disease activity score (NASH) [16] was used to evaluate the pathological changes in liver tissue. The MC group had a higher NASH score than the NC group (*p* < 0.05), and the MET, BU, OA, and three BOP groups all had lower NASH scores than the MC group (*p* < 0.05)—see Figure 4a,b.

### 3.4. Effects of BOP on Gut Microbiota

#### 3.4.1. Alpha and Beta Diversity Analysis

The Chao1, Shannon, Faith’s phylogenetic diversity, and Simpson indexes were used to estimate the alpha diversity of the rats (Table 3). However, no significant difference was found.

Weighted UniFrac distance was used to evaluate the beta diversity, and principal coordinate analysis (PCoA) was performed. According to the PCoA diagram (Figure 5), the samples in each group did not show obvious clustering. According to the PERMANOVA test, the NC and MC groups tended to have a difference in the weighted UniFrac distance (*p* = 0.056), but among the diabetic groups, there were no significant differences (*p* > 0.05), which was consistent with the PCoA result.

#### 3.4.2. Composition of Gut Microbiota

Figure 6a shows the relative abundance of the main 20 bacteria of each group at the genus level. Lactococcus was the most basic bacteria in most of the groups except for the NC, MC, and OA groups. All groups did not share the same top 3 bacteria, and the distribution seemed to be different among groups. To evaluate the differences, DeSeq2 analysis was conducted, and the results are shown in Figure 6b. Compared with the normal rats in the NC group, the diabetic rats in the MC group had a higher abundance of Weissella, Turicibacter, Eubacterium, Bacteroides, Enterococcus, and Dorea at the genus level; however, compared with the MC group, some of these bacteria tended to be lower in the intervention groups, such as Weissella, Turicibacter, Eubacterium, and Bacteroides. In addition, the BOP groups tended to have higher Lactobacillus and Phascolarctobacterium abundance than the MC group.

## 4. Discussion

Despite high evidence associated with the benefits of whole grains, whole grain consumption remains low globally. A study found that the mean consumption of whole grains was 38.4 g/day worldwide, far below the recommendation in many guidelines [17]. Many factors may prohibit the consumption of whole grain food, and bad taste and being hard to cook were found to be the two main reasons [18]. Some studies found that the tasting and processability could be improved just by mixing different grains together [11,12]. Many studies have contributed to the finding of high values of whole coarse cereals [7,19,20]; however, few studies have focused on the hypoglycemic effects of several coarse cereal mixtures [21]. Considering the synergetic or antagonistic effects of the phytochemicals, we studied a mixture of buckwheat:oats:peas at a 6:1:1 ratio in diabetic rats and found that BOP regulated the metabolism of glucose and lipids (although the differences in lipids levels between the BOP-M and MC group were not statistically significant), similar to the BU group. Moreover, the time to glucose peak in OGTT may be related to the metabolism of glucose and the risk of diabetes [22], and in our study, we found that diabetic rats had an early glucose peak. As the BOP groups and MET group had different peak times, they may have different mechanisms and synergetic effects, which needs to be studied.

To our surprise, the HDL-C in the MC group increased while interventions changed the increasing tendency. This may be related to the animal model. There is another study found increased HDL-C levels in a similar animal model [23]. HFD + streptozotocin was widely used to induce diabetes in rats, with stable hyperglycemia, similar metabolic characteristics, and natural history progression to human type 2 diabetes mellitus (T2DM) [24], but it may not be the same in every parameter. The HFD may contribute to the high HDL-C level, which needs to be studied further.

The liver plays an important role in the maintenance of glucose homeostasis. The relationship between liver injury and diabetes can be complicated. On the one hand, liver dysfunction may increase insulin resistance; on the other hand, diabetes may also cause liver damage through inflammation and oxidative stress. A study found that higher cyclic ALT and AST levels were positively correlated with the risk of T2DM, but reverse causality could not be proven [25]; therefore, BOP might play a hypoglycemic role by reducing liver injury.

Studies have found that intestinal bacteria play an important role in host health and are related to many chronic diseases, such as T2DM [26,27,28]. At present, people think intestinal bacteria may affect inflammation and glucose metabolism through short-chain fatty acids, bile acid, or endotoxin pathways. In T2DM patients, beneficial bacteria (such as Lactobacillus and Bifidobacterium) decreased, and harmful and conditional pathogenic bacteria (such as Enterococcus, Eubacterium, and Bacteroides) increased [28]. Although diabetes was considered to be associated with a reduction in microbial diversity, our study did not find a difference in alpha and beta diversity between normal and diabetic rats, which was similar to some studies [29,30]; however, the limited sample size may prevent the discovery of the relationship, and this needs to be studied further—some bacteria were however affected in this study. We found that BOP increased Lactobacillus, Eubacterium, and Phascolarctobacterium and decreased the abundance of Weissella, Turicibacter, and Bacteroides at the genus level. Phascolarctobacterium is a butyrate-producing bacterium. As a short-chain fatty acid, butyrate plays an important role in intestinal health. Turicibacter is associated with inflammation [31]. Because of the antimicrobial effects and potential cholesterol reduction effects of some species, Weissella is always considered to be beneficial. In our study, the diabetic rats had higher Weissella levels, and BOP decreased the levels. It was possible that some specific species played an important role in this process. Due to study limitations, we could not identify this possibility, and studies concentrating on the species or strain level may be helpful.

## 5. Conclusions

Our study demonstrated that the BOP diet had the effects of regulating lipid metabolism, decreasing liver injury, and changing the composition of intestinal bacteria in diabetics rats and may achieve hypoglycemic effects through these ways. BOP may be a potential stable food substitution.

## Figures and Tables

**Figure 1 foods-11-03938-f001:**
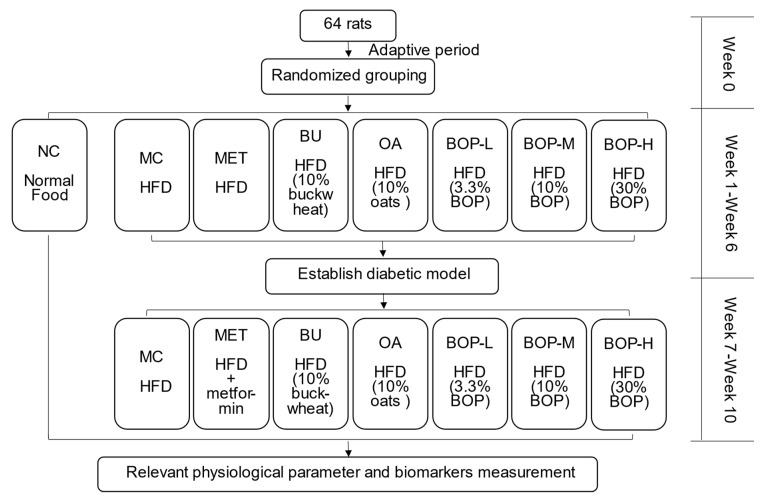
Schematic design of the study. NC: normal control group; MC: model control group; MET: metformin group; BU: buckwheat group; OA: oats group; BOP-L: low-dose BOP group; BOP-M: medium-dose BOP group; BOP-H: high-dose BOP group.

**Figure 2 foods-11-03938-f002:**
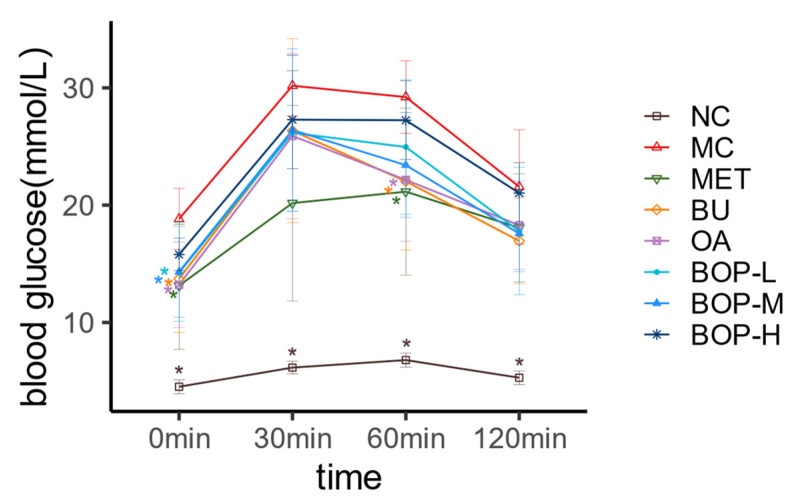
Effect of BOP on blood glucose change of oral glucose tolerance test (OGTT) at the 10th week (mean ± SD, *n* = 8). *: *p* < 0.05, compared with the MC group.

**Figure 3 foods-11-03938-f003:**
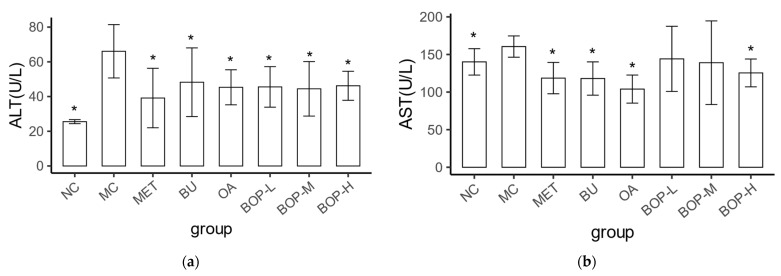
Effect of BOP on serum ALT (**a**) and AST (**b**) (mean ± SD, *n* = 8). *: *p* < 0.05, compared with the MC group.

**Figure 4 foods-11-03938-f004:**
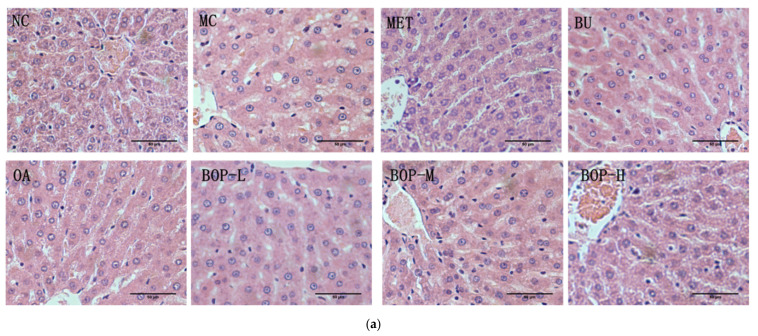
Effects of BOP on liver histology: (**a**) Representative microphotographs of liver sections, 200×; (**b**) NASH score, calculated by the three microphotographs each rat for three rats in each group; *: *p* < 0.05, compared with the MC group.

**Figure 5 foods-11-03938-f005:**
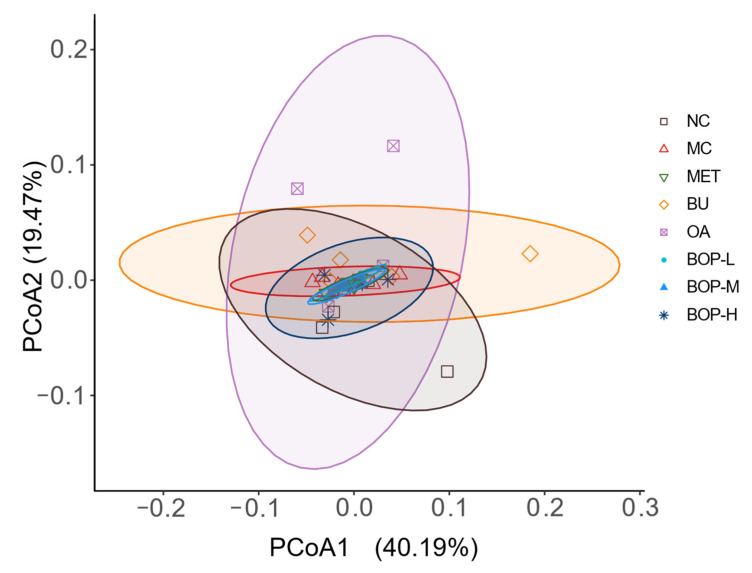
Effects of BOP on beta diversity (*n* = 5).

**Figure 6 foods-11-03938-f006:**
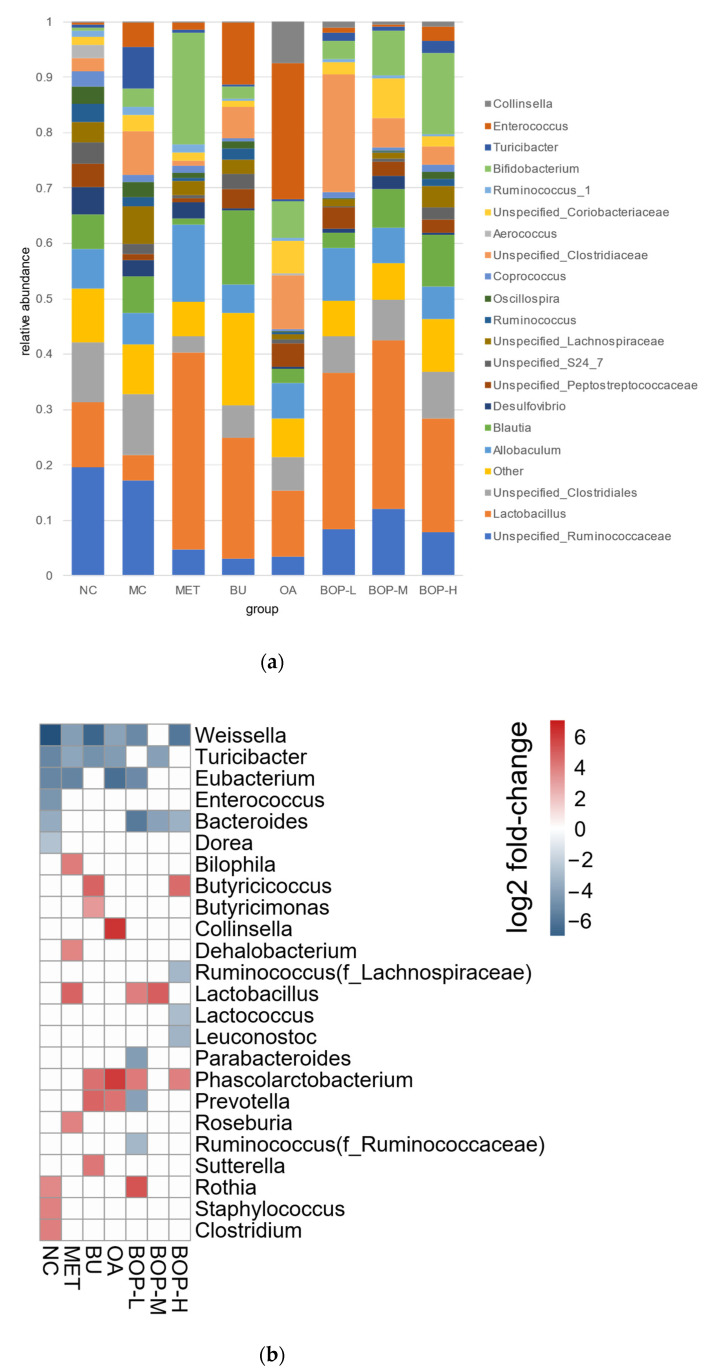
Composition of gut microbiota: (**a**) Relative abundance and composition of each group at the genus level; (**b**) results of the DeSeq2 analysis at the genus level (compared with the MC group) (*n* = 5).

**Table 1 foods-11-03938-t001:** Ingredient composition of experimental diets (g/kg).

	Normal Diet	HFD	Buckwheat Diet	Oats Diet	Low Dose BOP Diet	Medium Dose BOP Diet	High Dose BOP Diet
BOP	0.0	0.0	0.0	0.0	33.0	100.0	300.0
Buckwheat flour	0.0	0.0	100.0	0.0	0.0	0.0	0.0
Oats flour	0.0	0.0	0.0	100.0	0.0	0.0	0.0
Lactic casein	140.0	234.9	220.1	218.3	229.5	218.5	185.4
L-Cystine	1.8	2.1	2.1	2.1	2.1	2.1	2.1
Corn starch	465.7	85.5	12.0	18.9	62.0	14.3	0.0
Maltodextrin	155.0	117.5	117.5	117.5	117.5	117.5	0.0
Sucrose	100.0	207.7	207.6	207.7	207.7	207.7	197.1
Cellulose	50.0	58.7	51.1	51.4	56.0	50.7	34.9
Soybean Oil	40.0	29.4	29.4	29.4	29.4	29.4	29.4
Lard	0.0	208.5	204.5	199.0	207.1	204.1	195.4
Mineral Mix	35.0	41.1	41.1	41.1	41.1	41.1	41.1
Choline Bitartrate	2.5	2.9	2.9	2.9	2.9	2.9	2.9
Vitamin Mix	10.0	11.7	11.7	11.7	11.7	11.7	11.7
Tert-butylhydroquinone	0.01	0.01	0.01	0.01	0.01	0.01	0.01
Overall carbohydrate	770.7	469.4	468.7	468.7	468.8	468.2	466
Overall protein	141.8	237.0	236.2	236.4	236.9	236.6	235.5
Overall fat	40.0	237.9	237.2	237.5	237.7	237.3	236.2

HFD: high-fat diet; BOP: buckwheat–oat–pea composite flour.

**Table 2 foods-11-03938-t002:** Effects of BOP on blood lipids (mmol/L) (*n* = 6~8).

Groups	TC Median (25% IQR, 75% IQR)	TG Median (25% IQR, 75% IQR)	HDL-C Mean ± SD	LDL-C Median (25% IQR, 75% IQR)
NC	1.22 (1.12, 1.39) *	0.82 (0.60, 0.95) *	0.43 ± 0.05 *	0.27 (0.24, 0.28) *
MC	1.70 (1.43, 1.94)	1.68 (1.21, 2.35)	0.86 ± 0.17	0.32 (0.30, 0.46)
MET	1.41 (1.32, 1.46)	0.40 (0.36, 0.54) *	0.60 ± 0.06 *	0.28 (0.27, 0.32)
BU	1.17 (1.14, 1.49) *	0.26 (0.22, 1.29) *	0.65 ± 0.14 *	0.23 (0.18, 0.26)
OA	1.54 (1.44, 1.73)	0.70 (0.37, 1.14)	0.67 ± 0.15	0.33 (0.30, 0.47)
BOP-L	1.29 (1.15, 1.51) *	0.77 (0.51, 0.99)	0.65 ± 0.21 *	0.31 (0.26, 0.33)
BOP-M	1.73 (1.62, 1.82)	0.78 (0.48, 1.67)	0.75 ± 0.19	0.37 (0.31, 0.53)
BOP-H	1.38 (1.34, 1.51)	0.74 (0.64, 1.02)	0.68 ± 0.09	0.24 (0.23, 0.27)

*: *p* < 0.05, compared with the MC group.

**Table 3 foods-11-03938-t003:** Effects of BOP on alpha diversity (*n* = 5).

Group	Chao1 Median (25% IQR, 75% IQR)	Shannon Median (25% IQR, 75% IQR)	Faith’s Phylogenetics Diversity Median (25% IQR, 75% IQR)	Simpson Median (25% IQR, 75% IQR)
NC	255.0 (241.0, 270.0)	5.22 (4.85, 5.74)	17.87 (17.49, 18.07)	0.94 (0.91, 0.95)
MC	237.0 (176.0, 347.0)	5.68 (4.79, 6.34)	18.03 (12.02, 21.88)	0.96 (0.88, 0.96)
MET	187.0 (134.2, 292.0)	3.91 (3.07, 4.94)	11.46 (11.43, 13.52)	0.83 (0.74, 0.90)
BU	313.0 (217.0, 334.0)	4.83 (4.49, 4.96)	17.63 (11.76, 21.43)	0.92 (0.89, 0.93)
OA	165.0 (123.0, 225.0)	3.53 (3.47, 4.48)	13.51 (10.62, 18.27)	0.83 (0.72, 0.84)
BOP-L	147.0 (146.0, 153.0)	4.21 (3.97, 4.70)	11.93 (11.10, 12.22)	0.87 (0.85, 0.92)
BOP-M	184.0 (182.0, 204.1)	4.82 (4.37, 5.37)	14.51 (14.13, 16.23)	0.93 (0.90, 0.95)
BOP-H	288.2 (186.3, 453.0)	5.62 (5.05, 5.84)	14.07 (13.02, 18.57)	0.95 (0.94, 0.96)

## Data Availability

The data presented in this study are available on request from the corresponding author.

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
