# Peer review of "Hypoglycemic Effects and Mechanisms of Buckwheat–Oat–Pea Composite Flour in Diabetic Rats"

_foods, 2022, doi:10.3390/foods11233938_

Round 1

Reviewer 1 Report

The report to review the article entitled “Hypoglycemic effects of buckwheat-oat-pea composite flour in diabetic rats” from the Foods, MDPI, ISSN: 2304-8158. The paper was thoroughly checked and the following are my remarks regarding the status of the manuscript.

Abstract: this portion is though provided briefly and well explained however,

a.      There should be no use of headings such as; Background, methods, results and conclusion according to the standard format of the journal.

b.     The sentence in Line # 19 (i.e. and a diabetic rat model was used) is irrelevant in the background section and may be excluded.

c.      The sentence in line # 27 may be written as; the outcomes of the current study revealed that BOP is a potential stable food substitution.

Introduction

a.      There is often no space after the completion of the sentence and before the citation of reference (Line # 36, 39, 42, 43, 45, 48, 49, 212, 259, 261, 272, 282, 289, 292 and 298).

b.     According to the standard format of the journal the citation of reference must be numbered. The citation by name may therefore be replaced with number in line # 48 and 49.

c.      The author have stated the outcomes of previous study by which author mean the outcomes of an unpublished material in line # 53 to 59 (i.e. In our previous study, we found that ... and the 6:1:1 pattern was found) how is it logical? This may reflect to the audience that the manuscript is incomplete or it may be the part of large data merely managed to increase the number of publications. The author is advised to include the same data (if already performed) in this paper to authenticate the work done.  

·       Material and methods section do not provides details of some aspects such as;

a.      Italicized biological names in line # 65, 66

b.     Explain how the ratio of 6:1:1 were prepared (If weight or volume were used what were the units?)

c.      There is used of some abbreviations which are not explained such as; AIN-93 M,

d.     Section 2.5 don not explain how TC, TG, LDL-C, HDL-C, ALT and AST in serum were detected at least this section may provided with brief methodology for the said analysis. Section 2.5 and 2.6 why not provided with reference of any protocol/methodology.

e.      Section 2.7 state the amplification of genes using PCR however this section is not provided with the details of PCR conditions and enzymes used etc.

f.      Section 2.7 is provided with statistical analysis used. For avoiding the confusion of audience it is better;

i.                 To separate the statistical analysis from section 2.7 and merged it with section 2.8 or

ii.               To state statistical analysis along with each experiment.

g.     The authors mentioned the use of one-way ANOVA check for normality of distributions. Prior to the use of a post hoc test, you should use Shapiro-Wilks test to check out the normality of data followed by ANOVA and then by one of the post hoc test for determining specific group differences.

·       Results section have been described in details however;

a.      For consistency and attraction of the readers only significant findings may be explained as the results in the table and figures are self explanatory.

b.     There is often no space in the median and parentheses consisting IQR values in table 1 line # 194.

c.      There is often no space in the values and values in parentheses in table 2 line # 228.

·       Discussion section

a.      This section needs updated references.

b.     There is often no space after the completion of the sentence and before the citation of reference such as line # 259, 261, 282, 289, 298.

c.      Citation “Ruggiero E” in line # 260 does not follow the standard format of the journal.

d.     There is a continuous use of some studies and few studies without any proper reference from line # 262 to 269 which is highly unprofessional for a scientific writing. Update this section with proper references.

e.      Line # 271 and 272 does not follow the journal standard format (i.e. Guo X-X and her colleagues) and as scientist how gender (her/his) of author from an article may be known to someone out of the certain ethnic groups/geographical region? While, scientific writing you must avoid such sentences.

f.      Again use of word “many studies” in line # 284; can you mention any one reference please?

g.     Overall discussion need some important references as some aspects have not been described with proper citations.

·       Conclusion

a.      The conclusion line 307 i.e. “and the intestinal bacterial structure was changed in vivo” is in contrast with the discussion line # 290 to line # 292 (i.e. “our study did not find a difference in alpha and beta diversity between normal and diabetic rats, which was similar to some studies [18, 19]”). How two different stance can be justified explain please?

b.     The whole conclusion section needs to be rephrasing with future recommendations.

General remarks and decision:

This paper describes the in-vivo hypoglycemic effects of buckwheat, oats, and peas (6:1:1) using animal models (rats) and its underlined mechanism. The work done falls within the scope of Foods. However, the manuscript possesses some major issues given in detail. Above being the position of the manuscript, I, therefore, suggest the "major review" of the article.

Reviewer 2 Report

The authors raise an important point; specifically, the universal low consumption of whole grains across the globe. The study evaluates the impact of a series of unique combinations of whole grains on blood glucose levels. The work provides important insights into the role of those macro and micronutrients on insulin control. The authors also offer valuable findings on contributing pathological considerations to blood glucose control.

The Methods need to be reworked and better organized so they more fully describe the study. It might be helpful to mirror the organization used in the Discussion.

Throughout add period after “al” in et al.

Author affiliations: Only include emails, organizational information needs to be added.

Line 32: Add citation.

Line 33: Add citation.

Lines 53-54: Recommend changing “In our a previous study, we found that a mixture of buckwheat, oats and peas (weight 53 ratio= 6:1:1) could decreased insulin resistance in vitro (not published).”   Also, use unpublished research citation style and cite this reference.

Line 61: “and explore the underlying mechanisms”  recommend adding after mechanisms, including lipid metabolism, liver functioning, and gut microbiota.

Line 69: Delete “Basically”

2.1.2 Recommend discussing how these products were used in the various diets.

2.1  Recommend adding a table that shows macronutrient compositions of different variations of the diets. Also include amounts of other products described in 2.1.2.

Line 128: Define GAUC, 3-hour postprandial glucose excursion?

2.5 Define acronyms.

Line 260: Delete “E”.

Line 263: Add citation.

Line 264: Add citation.

Line 267: Delete “X-X”.

Line 273: “was widely used”  Did Guo report on these other studies or are additional references needed to support this sentence? Clarification is needed.

Line 280: Add citation.

Round 2

Reviewer 1 Report

The changes made by the author according to the comments are appreciated.

Author Response

Thank you for your approval. We really appreciate your time and all the helpful comments. 

Reviewer 2 Report

The edits made by the authors addressed my concerns. I recommend publishing this manuscript.

Author Response

We really appreciate your careful review and helpful comments. Thank you for your recommendation for publication.